# A prognostic model for predicting the duration of 20,049 sickness absence spells due to shoulder lesions in a population-based cohort in Sweden

**Katalin Gémes**[1]*, **Johanna Holm**[1], **Paolo Frumento**[2], **Gino Almondo**[1], **Matteo Bottai**[2], **Emilie Friberg**[1], **Kristina Alexanderson**[1]

**1** Division of Insurance Medicine, Department of Clinical Neuroscience, Karolinska Institutet, Stockholm, Sweden, **2** Division of Biostatistics, Institute of Environmental Medicine, Karolinska Institutet, Stockholm, Sweden

* katalin.gemes@ki.se

## Abstract

### Main objective

Sickness absence duration for shoulder lesion patients is difficult to prognosticate, and scientific evidence for the sick-listing practice is lacking. Our objective was to develop a clinically implementable prediction model for the duration of a sickness absence spell due to shoulder lesions.

### Methods

All new sickness absence spells due to shoulder lesions (ICD-10-code: M75) issued in the period January 2010—June 2012 that were longer than 14 days were identified through the nationwide sickness absence insurance register. Information on predictors was linked from four other nationwide registers. Piecewise-constant hazards regression models were fitted to predict duration of sickness absence. The model was developed and validated using split sample validation. Variable selection was based on log-likelihood loss ranking when excluding a variable from the model. The model was evaluated using calibration plots and the *c*-statistic.

### Results

20 049 sickness absence spells were identified, of which 34% lasted >90 days. Predictors included in the model were age, sex, geographical region, occupational status, educational level, birth country, specialized healthcare at start of the spell, number of sickness absence days in the last 12 months, and specialized healthcare the last 12 months, before start date of the index sickness absence spell. The model was satisfactorily specified and calibrated. Overall *c*-statistic was 0.54 (95% CI 0.53–0.55). *C*-statistic for predicting durations >90, >180, and >365 days was 0.61, 0.66, and 0.74, respectively.

**Data Availability Statement:** The data used in this study is administered by the Division of Insurance Medicine, Karolinska Institutet, and cannot be

made public. According to the General Data Protection Regulation, the Swedish law SFS 2018:218, the Swedish Data Protection Act, the Swedish Ethical Review Act, and the Public Access to Information and Secrecy Act, these types of sensitive data can only be made available, after legal review, for researchers who meet the criteria for access to this type of sensitive and confidential data. Readers may contact Professor and Head of Division Ellenor Mittendorfer-Rutz (ellenor. mittendorfer-rutz@ki.se) regarding the data.

**Funding:** This work was financially supported by a research grant from the Social Insurance Agency [grant number 059159-2015, receiver: KA]. We utilised data from the REWHARD consortium supported by the Swedish Research Council (grant number 2017-00624, receiver: KA). There was no additional external funding received for this study. The funders had no role in study design, data collection and analysis, decision to publish, or preparation of the manuscript.

**Competing interests:** The authors have declared that no competing interests exist.

## Significance

The model can be used to predict the duration of sickness absence due to shoulder lesions. Covariates had limited predictive power but could discriminate the very long sickness absence spells from the rest.

## Introduction

Shoulder lesions are the third most common group among the musculoskeletal disorders [1], and musculoskeletal disorders comprise the most or second most common sickness absence (SA) diagnoses [2]. The clinical course is heterogeneous with a subgroup of patients experiencing long-term SA, and the prognostication of a newly diagnosed case is challenging [1, 3–5]. Physicians issuing SA certificates due to shoulder lesion are expected to estimate the needed duration of SA, despite lack of scientific knowledge to base this decision on [6–9]. Physicians have consequently been found to be inaccurate in prognosticating duration of SA [10–12], especially for long-lasting SA due to musculoskeletal disorders [10, 13]. Prediction models for estimating the duration of SA could thus potentially be very useful in a clinical setting and give hints in for which cases additional measures, such as occupational rehabilitation measures, might be needed. However, knowledge of factors predicting the duration of SA due to shoulder lesions is scarce [6, 8, 14]. Most previous studies on prediction models for SA due to shoulder lesion have focused on predicting incidence of SA, not the duration thereof [6, 15, 16] or the probability of getting back to work after SA [8]. Moreover, these few studies were based on small samples, that is, studies based on large, non-selected data are needed. Physicians in primary healthcare, that is, general practitioners, have asked for a soft-ware based tool regarding such predictions, not requiring much work or time to use. Therefore, the aim of this study was to develop a parsimonious prediction model for duration of SA due to shoulder lesion, which can easily be implemented in clinical practice.

## Methods

The reporting of this study was conducted according to the TRIPOD statement for transparent reporting of prediction models [17] and STROBE guidelines [18].

A model was developed and internally validated in a Swedish population-based study, including all SA spells that lasted >14 days and that begun during the 2.5-year period 1 January 2010–30 June 2012 and were due to due to shoulder lesion as the main diagnosis (ICD-10 code: M75) [19] among individuals aged 18–64 at the beginning of the spell. The SA spells were followed until they ended or became >1000 days long. The SA spells were identified from the micro-data for analysis of the social insurance system (MiDAS) held by the Swedish Social Insurance Agency (SSIA), a nationwide register holding information on dates, extent (full- or part-time), and main diagnosis of reimbursed SA spells and disability pensions [20].

Microdata from the following four other population-based nationwide administrative registers in Sweden were linked at individual level based on the unique identification number of all residents in Sweden [21]: From Statistics Sweden, information on sociodemographics (age, sex, country of birth, educational level, employment status, occupational sector, family situation, and geographical area) was obtained from the longitudinal integration database for health insurance and labor market studies (LISA) [22]. From the National Board of Health and Welfare, the in- and specialized outpatient registers [23], the prescribed drug register [24], and the

cause of death register [25] were used to obtain information on history of secondary health-care, drug dispenses, and date of death.

## Sickness absence and disability pension benefits in Sweden

In Sweden, all residents aged 16 and above with an income from work or unemployment bene-fit are covered by the public SA insurance and can claim SA benefits if their work capacity is reduced due to disease or injury [26]. Day 1 is a waiting day, with 100% loss of income. After 7 days, a medical certificate from the treating physician is required. The employer reimburses income loss during days 2–14, after which SA benefits are administered by the SSIA, for unem-ployed this happens from day 2. In order not to introduce bias regarding unemployment, we only included SA spells >14 days. This also led to high validity of SA diagnoses. There is no upper limit of duration of SA spells. Additionally, all residents aged 19–64 can be granted dis-ability pension (DP) if they have long-term or permanently reduced work incapacity due to disease or injury. Both SA and DP benefits can be granted for full-time (100%) or part-time (75%, 50%, or 25%) of ordinary work hours. SA benefits cover 80% and DP benefits cover 64% of lost income, up to a certain level.

## Outcome

The main outcome variable was SA duration in days, defined as the total number of days from the start date of the SA spell to the last day with SA benefits, excluding days for which no SA benefit was granted from SSIA (e.g., during planned vacation). Sickness absence spells that occurred within five days of the end of a previous SA spell are automatically merged and defined as one SA spell in the MiDAS database, as in such cases individuals are entitled to SA benefit from the SSIA from day one of the new SA spell. For the purpose of our analysis, we capped 297 spells lasting more than 999 days at 1000 days, to control for extreme outliers.

## Predictors

A priori, it was decided to develop a final model with no more than nine predictors, including age, sex, and geographic region, which can be easily and reliably assessed in a short time period during a primary healthcare visit to promote implementation of the model into clinical prac-tice. Initially, around 130 predictors were considered to be included in the model. In order to decrease the number of predictors to a reasonable number that could be used to build the pre-diction model, several parallel preselection processes were applied, based on feasibility for clin-ical implementation and scope for association to the outcome. The majority of the initial predictors were redundant or partially redundant and/or highly collinear with at least one other predictor, e.g., net and gross days of the same measures in days. Therefore, based on analyses results, a subset of them were selected for further consideration. For predictors that were hierarchically related, e.g., multi-morbidity and cause-specific morbidity, general and cause-specific hospitalization, etc., choosing only the general/coarse variables over the specific ones would improve both parsimony and the scope for implementation of the model in clinical settings and was, therefore, the selected strategy. This variable selection procedure resulted in a set of 14 easily available and easy to assess variables, with minimal collinearity and strong asso-ciations with the outcome for modeling: age (categorized into 18–34, 35–40, 41–50, 51–57, 58–64 years), sex (women/men); geographical region (categorized into the five groups: North, Mid, West, South Sweden, and Stockholm/Gotland); educational level (elementary (≤9 years), high school (10–12 years), and university/college (>12 years)); family situation (composite four-level variable of 'married/cohabitant' yes/no, and 'living with children <18 years old' yes/no); country of birth ("Sweden", "Non-Swedish Nordic country", "Non-Nordic EU country",

"Non-EU country"); number of gross days with SA benefits during the 12 months preceding the start date of the SA spell (0, 1–90, 91–180, or >180 days); number of specialized outpatient healthcare visits during the 12 months preceding start of the SA spell (0, 1–2, >2) (based on the median among those with such visits); number of inpatient days the 12 months preceding start of the SA spell (0, 1–2, >2) (based on the median number among those hospitalized, n = 2); extent of SA benefit at initiation of spell (25%, 50%, 75%, or 100%); partial DP at start of SA spell (yes/no); employment status at start of SA spell ("employed/Student", "parental leave", "unemployed"), multi-morbidity (defined as taking out >1 drug dispenses of at least 3 different anatomical chemical classification (ATC) codes (at 1-digit level) during the 12 months preceding start date of the SA spell: yes/no); and secondary healthcare at SA initiation (yes/no). In calculation of healthcare visits and hospitalization days, we excluded healthcare visits due to full-term uncomplicated deliveries (ICD-10: O80), counseling, general medical advice and screening (ICD-10 codes: Z00-Z99, except for Z73.0 coding for 'burnout syndrome' which was included). Information was complete for all variables, with the exceptions of 0.05% of SA spells missing information on country of birth (these were coded as "non-EU country") and 0.3% of spells missing information on educational level (coded as lowest level of education). (Some of the variables that in the initial process did not show strong enough predictive value to be included among those 14, were two of the work related: namely type of sector /(private vs. public) and type of work (blue vs. white collar)).

## Statistical analysis

Piecewise-constant hazards models were fitted to predict the duration of the SA spell, using the 'pch' package version 1.3 [27]. Piecewise-constant hazards models allow both the baseline hazard of the event and the hazard associated with the predictors to vary over time, by fitting separate constant hazard (exponential) models for each pre-defined time interval using Poisson regression. The model was specified to contain 20 piecewise intervals.

Internal validation was performed by split-sample validation, splitting the data into development (70%) and validation (30%) data, using random sampling without replacement. Variable selection was performed in the development set, and the performance of the model was evaluated by predicting the duration of SA on observations in the validation data. The validation data was not used during any part of the model building. Variable selection was performed by ranking the log-likelihood loss for each variable when excluding it from the model. The six predictors with highest log-likelihood loss (after excluding age, sex, and region, which were forced into the model) were retained in the final model.

## Formal evaluation of the model

Goodness-of-fit was assessed using Akaike's and Bayesian information criteria. Specification of the model was assessed in the validation data with a quantile-quantile plot of survival probabilities at observed time-of-event T. The overall discriminatory capacity was assessed in the validation data using Harrell's $c$-statistic [28]; the proportion of paired subjects where the ranking of the predicted and observed survival times were in agreement [29]. Confidence intervals (CI) for $c$ were obtained using bootstrap with 1000 times resampling. The discriminatory accuracy for three binary outcomes of predicting duration longer than 90, 180, or 365 days, respectively, was assessed using receiver operating characteristics (ROC) curves and their corresponding $c$-statistic (technically, the AUC: for binary outcomes AUC corresponds to $c$), with CI's calculated from bootstrap resampling [30]. Box plots of predicted probabilities with and without the respective binary outcome were plotted. Calibration for each of the three

binary outcomes was assessed visually by plotting the smoothed calibration curve using natural splines with 3 degrees of freedom.

All statistical analysis was done in R version 3.4.3 (RCoreTeam, 2017) (packages 'pch', 'pROC', 'e1071', ´Hmisc', 'ggplot2' [27, 30, 31]).

The project was approved by the Regional Ethical Review Board of Stockholm (DNR 2007/ 762-31 and 2016/1533-32) and the study was carried out in accordance with the Declaration of Helsinki [32]. The Review Board of Stockholm waived the need for informed consent [32, 33].

## Results

A total of 20,049 new SA spells due to shoulder lesion, from 17,970 individuals, were identified for the 2.5-year period and included in the analysis. Their characteristics are tabulated in Table 1, separately for the development (n = 14,034 SA spells) and validation (n = 6015 SA spells) data.

The distribution of the duration of the SA spells was heavily positively skewed, with a group of outliers of very long SA spells (Fig 1). The mode of SA duration in the development data was 21 days, with a median of 58 days and interquartile range 30 to 129 days. The distribution showed the same characteristics in the validation data, with a slightly narrower interquartile range (29 to 124 days) (Table 1).

Model predictors tabulated in the development and validation dataset are shown in Table 2. Predictors age, sex, geographical region, secondary care at baseline, occupational status, educational level, country of birth, history of SA benefits, and history of specialized outpatient visits in the 12-month period before start of the SA spell were the nine predictors showing greatest log-likelihood loss and were therefore selected as the final model.

### Predictive performance of the model

Both AIC and BIC were slightly lower in the final rather than the full model, indicating a better fit (Table 3). The final model was well calibrated and correctly specified as the probability integral transform value of the model was uniform (Fig 2). The overall discriminatory capacity on individual level was poor, with $c$ = 0.54 (95% CI 0.53–0.54) (Table 3). Binary predictions of risk of long-term SA duration showed poor performance for short-term outcomes but good discriminatory ability for predicting long durations of SA with $c$-statistic = 0.61 (95% CI 0.59–

**Table 1. Descriptive statistics of all the 14034 included incident sickness absence (SA) spells due to M75.**

| Characteristics | Development data | Validation data |
|---|---|---|
| Number of SA spells | 14034 | 6015 |
| Number of unique individuals | 12985 | 5829 |
| Number of individuals with more than one SA spell | 978 | 180 |
| Median duration of SA spell (IQR) | 58 (30–129) | 57 (29–124) |
| Mean duration of SA spells | 125 | 125 |
| Mode duration of SA spells | 21 | 21 |
| Duration of SA spell (n, %) | | |
| 15–90 days | 9154 (65) | 4021 (67) |
| 91–180 days | 2406 (17) | (17) |
| 181–365 days | 1529 (11) | (10) |
| >365 days | 945 (7) | 411 (7) |
| ≥1000 days | 208 (2) | 89 (2) |

IQR = interquartile range.

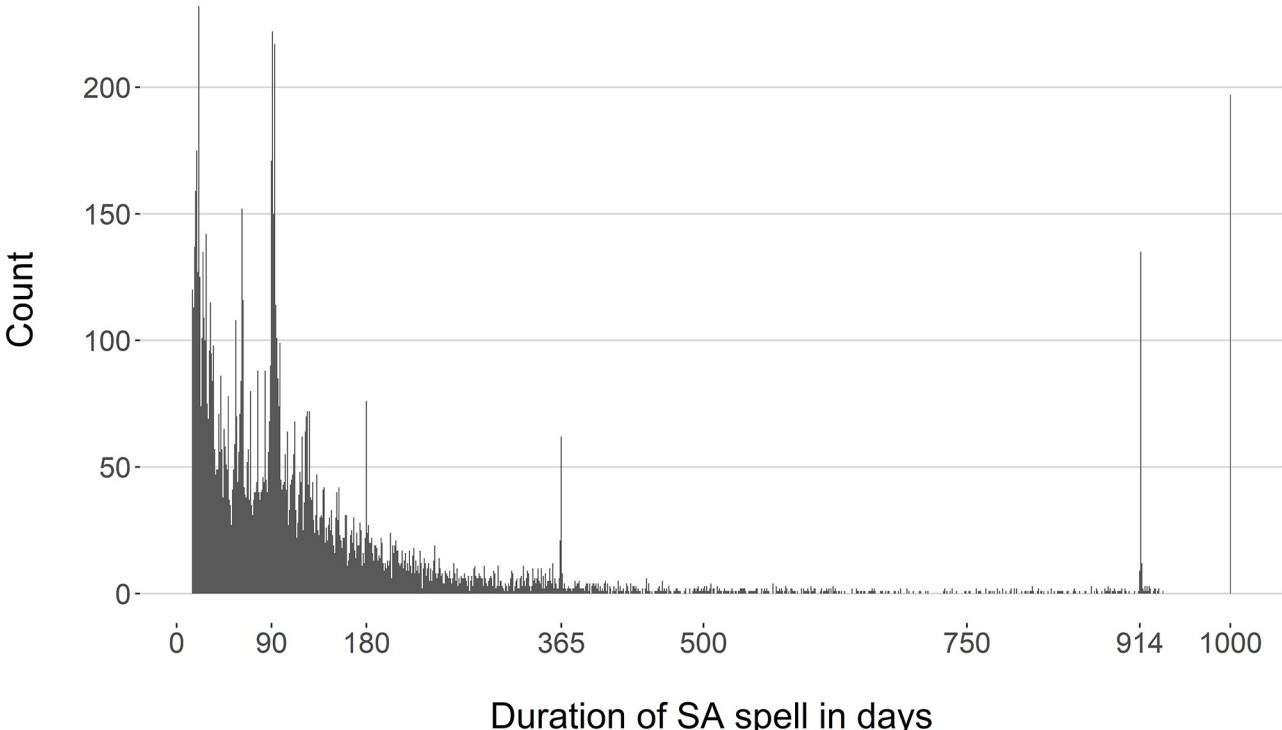

**Fig 1. Histogram of the distribution of sickness absence spells (SA spells).**

0.62) for >90 days, 0.66 (0.64–0.68) for >180 days, and 0.74 (0.71–0.77) for cutoff at >365 days (Table 3, Fig 3). On the contrary, the model was well calibrated for the binary 90-day outcome, but less well calibrated for the long-term predictions at 180 and 365 days (S1 Fig).

## Discussion

We developed a prediction model that could be implemented in primary healthcare. The model was based on information about all new 20,049 SA spells due to shoulder lesion >14 days that occurred in Sweden during a 2.5-year period. The model was correctly specified but it had poor overall discriminatory capacity (*c*-statistic = 0.54). Dichotomizing the outcome to predictions of SA beyond >90, 180, or 365 days, respectively, showed poor discrimination for the 90-day outcome (*c* = 0.61) but good discrimination for predicting being on SA >365 days (*c* = 0.74); it is probably that the heavy skew of the distribution of duration of SA spells contributed to the poor overall *c-statistics* despite the good discriminatory ability for longer durations. The calibration was, however, satisfactory making this a useful model for prognosticating the distributions of duration of SA in a new, similar population. However, before applying this model to external settings it would be highly recommended to validate the model externally.

The individual level prognostication is hampered by the limited discriminatory ability, reflecting the stochastic nature of the underlying processes and the lack of knowledge about potentially causal determinants of the duration of SA spells. It is possible that adding more predictors would have improved model performance, however, our aim was to balance a parsimonious model that could easily be implemented in daily clinical practice. A more desirable strategy than adding many more predictors would be to identify better predictors and make use of those. For such future model improvements, more information is needed on factors associated with duration of SA due to shoulder lesion as the literature is scarce [6, 8].

**Table 2. Tabulation of baseline predictors for the development and validation data sets.**

| Predictor | Development data (n spells, %) | Validation data (n spells, %) |
|---|---|---|
|  | 14034 (100) | 6015 (100) |
| Sex |  |  |
| Women | 7268 (52) | 3137 (52) |
| Men | 6766 (48) | 2878 (48) |
| Age at start of sickness absence spell |  |  |
| 18–30 years | 583 (4) | 265 (4) |
| 31–40 years | 1699 (12) | 710 (12) |
| 41–50 years | 4411 (31) | 1849 (31) |
| 51–57 years | 3937 (28) | 1711 (28) |
| 58–64 years | 3414 (24) | 1480 (25) |
| Geographical region |  |  |
| North | 1827 (13) | 802 (13) |
| Middle | 2064 (15) | 845 (14) |
| Stockholm/Gotland | 2640 (19) | 1146 (19) |
| West | 4466 (32) | 1902 (32) |
| South | 3037 (22) | 1320 (22) |
| Educational level (years) |  |  |
| Elementary school (≤9 years) | 2988 (21) | 1310 (22) |
| High School (10–12 years) | 8590 (61) | 3659 (61) |
| College/university (>12 years) | 2456 (18) | 1046 (17) |
| Country of birth |  |  |
| Sweden | 11391 (81) | 4900 (81) |
| Non-Swedish Nordic country | 680 (5) | 296 (5) |
| Non-Nordic European Union country | 309 (2) | 126 (2) |
| Non-EU country | 1654 (12) | 693 (12) |
| Number sickness absence days in the previous 12 months |  |  |
| 0 | 19647 (69) | 4125 (69) |
| 1–90 | 3306 (24) | 1438 (24) |
| 91–180 | 635 (5) | 256 (4) |
| 181–365 | 446 (3) | 196 (3) |
| Specialized outpatient care visits[1] in the previous 12 months |  |  |
| 0 | 5058 (36) | 2213 (37) |
| 1–2 | 5255 (37) | 2258 (38) |
| >2 | 3721 (27) | 1544 (26) |
| Employment status at start of sickness absence spell |  |  |
| Employed | 13275 (95) | 5699 (95) |
| Parental leave | 63 (<0) | 19 (<0) |
| Student | 4 (<0) | 1 (<0) |
| Unemployed | 692 (5) | 296 (5) |
| Specialized healthcare at start of sickness absence spell |  |  |
| No | 8546 (61) | 3656 (61) |
| Yes | 5488 (39) | 2359 (39) |

[1] = Excluding O80, and Z-codes Z00-Z99 except Z73.0.

**Table 3. Model fit and model performance.**

| | Full model (14 predictors) | Final model (9 predictors) |
|---|---|---|
| Model fit *(Development data, n = 14034 spells)* | | |
| N free parameters | 640 | 460 |
| Log-likelihood | -75824 | -75951 |
| AIC | 152929 | 152823 |
| BIC | 157760 | 156296 |
| Model performance *(Validation data, n = 6015 spells)* | | |
| *c*-statistic overall | NA | 0.54 (0.53–0.55) |
| *c* P(T >90 days) | NA | 0.61 (0.59–0.62) |
| *c* P(T >180 days) | NA | 0.66 (0.64–0.68) |
| *c* P(T >365 days) | NA | 0.74 (0.71–0.77) |

To the best of our knowledge, there have not been any previous models developed for predicting risk of a SA spell due to shoulder lesions becoming very long and our results are, therefore, difficult to compare to the literature. The most relevant studies to compare our results to are thus either studies concerned with predicting incidence of long term SA due to any diagnoses [34], or small studies that predicted 2-year work participation among sickness absentees due to neck and shoulder pain using sociodemographic and patient-reported variables [8] or duration of SA among hospitalized patients undergoing shoulder arthroscopy within a specific

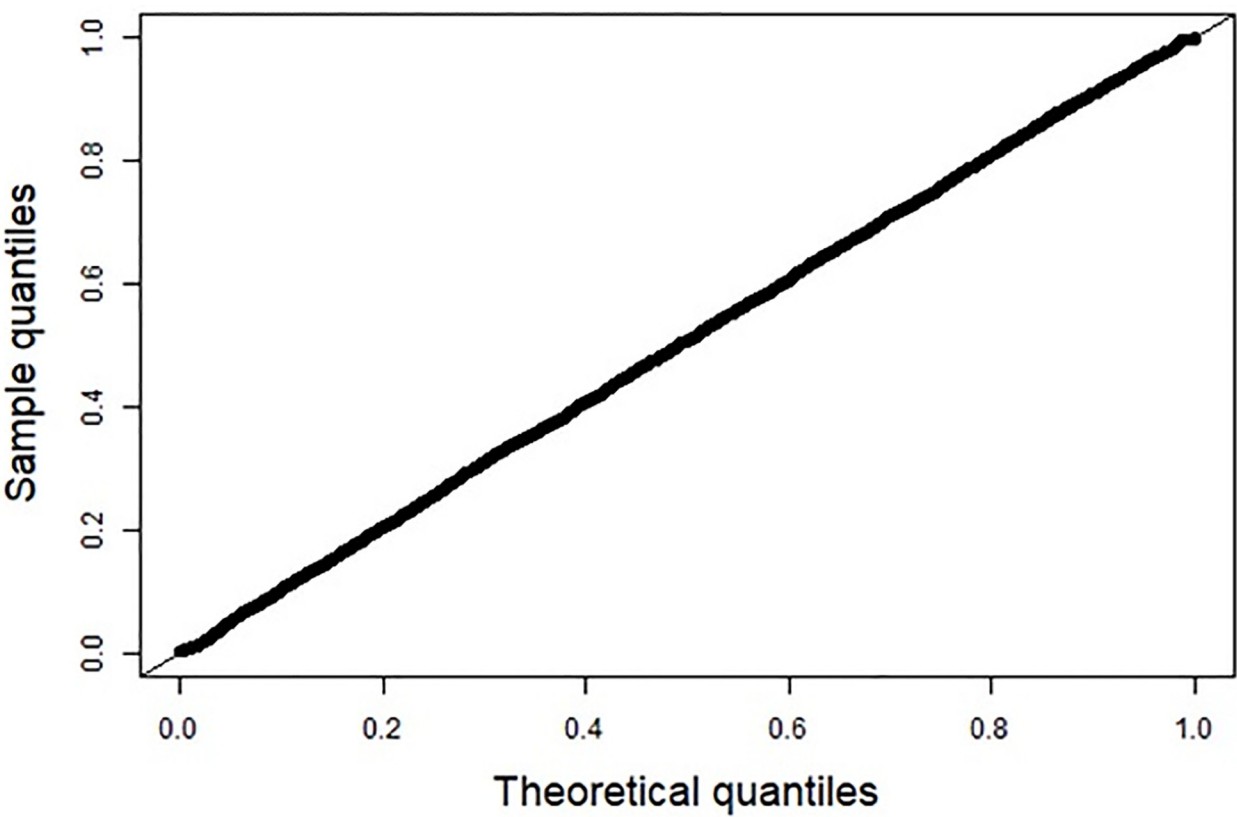

**Fig 2. Quantile-quantile plot of model calibration: Survival probability at observed time of event.**

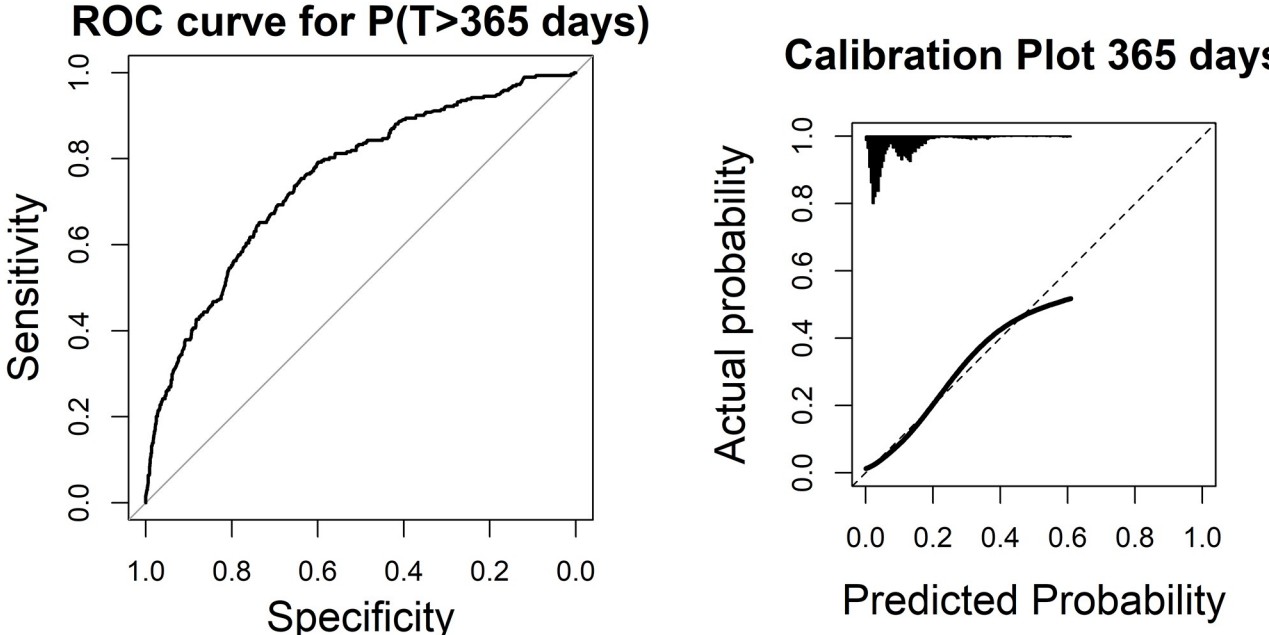

**Fig 3. Receiver-operating characteristics (ROC) curves and boxplots of predicted duration of a sickness absence (SA) spell for outcomes A; SA >90, B; SA >180, and C; SA >365 days, respectively.** The corresponding *c*-statistic is defined as the area under the ROC curve (AUC). The diagonal line in the ROC curve indicates a *c*-statistic of 0.5, no better discrimination than random chance.

insurance company [14]. Moll et al. [8] found that age, sex, education, sick leave duration and von Knoch et al. [14] identified age and pre-surgical SA to be (univariate) as important predictors of work participation and duration of SA post arthroscopic shoulder surgery. Similarly, we found age and history of SA the year preceding start of the spell to add information to our model. More generally, age and history of SA have also been identified as predictors of the risk of long-term (>90 days) SA in a recent Finnish cohort study, although this study focused on the incidence, and not duration, of SA spells due to any diagnoses in a cohort of employees [34]. The lack of studies assessing the duration of SA due to shoulder lesions and its predictors, highlights the relevance of our current work but should also be an encouragement to increase research efforts in this field. As SA certification is by no means an uncommon or unproblematic task for physicians [7, 9, 35], the lack of scientific evidence on which to base this practice on is perhaps particularly concerning. This prediction model can be used in Sweden, during primary healthcare consultations when patients begin a SA spell due to shoulder lesion. The general practitioner inserts information on the relevant predictors about the patient and optionally specifies the length of SA for which to obtain a probability. The model outputs a probability score that can be used by the practitioner to early identify patients with high risk for long-term SA. The model's output can also be used as a basis for discussion with the patient regarding their current SA. Early identification of those SA spells that are at high risk of becoming long means that resources and support, such as rehabilitation measures, can more accurately be targeted to those with greater need of them.

The main strength of our work is the population-based design with complete coverage of all SA spells >14 days in a whole country, linked to high quality information from several nationwide registers [21–23, 25], carried out in a population with high employment frequency among both women and men and in higher ages. Thus, effects of healthy worker selection are limited. Additionally, the material was large enough to allow for both a large development and

internal validation sample. A strength of the resulting model is the choice of predictors that are pragmatic to assess and clinically easy to implement. The current work is, however, also limited by the registry-based approach in some areas. We had no information on disease severity or on type of treatment (e.g., surgery or not), nor information on specific SA diagnostic M75 sub groups (we only had information about the three-digit level ICD-10 code from MiDAS). Such types of information could have improved the discriminatory ability of the model and would be preferable to include in future studies [6]. Nor did we have information on individual lifestyle factors, which also might have added predictive power to the model [34].

Nevertheless, the model is still valid for making prognostications in the setting of initiating a SA spell, which is how it is intended to be used. Furthermore, our results concern SA spells initiated in 2010–2012, but as not major legislative changes regarding SA benefit have been implemented in 2012–2020, the results are likely to be valid to later periods. However, the predictive models need to be updated to examine possible needs of changing predictors—when new data is available to ensure its validity during later periods. Currently, the model is only validated internally under the conditions applying to the Swedish healthcare and social insurance system, and may only generalize to similar settings in terms of employment frequencies SA benefit and disability pension coverage [36].

In Sweden, several nationwide interventions have been introduced in the decade before those SA spells begun, such as interventions to increase competence in insurance medicine among physicians and social insurance staff, stricter time lines regarding when to assess claimants right to prolonged the SA spell (e.g., at day 90, 180, 365, and 914, as can be seen in Fig 1), and in 2007, recommendations from the Board of Health and Welfare regarding sickness certification and duration of SA spells due to specific diagnoses, including M75, namely due to M75.0 (frozen shoulder), M75.1 (rotator cuff injuries), M75.3 (calcific tendinitis), and M75.4 (impingement) [37, 38]. The degree to which those interventions might have had an impact on how long the SA spells became would require other types of studies and data.

## Conclusions

A prediction model of the duration of sickness absence (SA) spells due to shoulder lesion was developed, with ease of clinical implementation in mind. Discriminatory ability was poor in the short term but improved for predicting very long durations of SA. The model is useful for prognostications of duration of SA spells and can provide information regarding which cases that might be in need of extra rehabilitation measures to promote return to work or other activity. However, future external validation is recommended.

## Supporting information

**S1 Fig. Calibration plots of binary outcomes SA > 90, 180 and 365 days respectively.** (DOCX)

## Author Contributions

**Conceptualization:** Johanna Holm, Emilie Friberg, Kristina Alexanderson.

**Data curation:** Kristina Alexanderson.

**Formal analysis:** Katalin Gémes, Johanna Holm, Paolo Frumento, Gino Almondo, Matteo Bottai.

**Funding acquisition:** Emilie Friberg, Kristina Alexanderson.

**Methodology:** Paolo Frumento, Gino Almondo, Matteo Bottai, Emilie Friberg, Kristina Alexanderson.

**Project administration:** Kristina Alexanderson.

**Resources:** Kristina Alexanderson.

**Software:** Paolo Frumento, Gino Almondo, Matteo Bottai.

**Supervision:** Kristina Alexanderson.

**Writing – original draft:** Katalin Gémes, Johanna Holm.

**Writing – review & editing:** Katalin Gémes, Johanna Holm, Paolo Frumento, Gino Almondo, Matteo Bottai, Emilie Friberg, Kristina Alexanderson.

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
