## [Decision Letter · Decision Letter 0]

16 Aug 2022

PONE-D-22-17574A prognostic model for predicting the duration of 20,049 sickness absence spells due to shoulder lesions in a population-based cohort in SwedenPLOS ONE

Dear Dr. Gemes,

Thank you for submitting your manuscript to PLOS ONE. After careful consideration, we feel that it has merit but does not fully meet PLOS ONE’s publication criteria as it currently stands. Therefore, we invite you to submit a revised version of the manuscript that addresses the points raised during the review process.

Please note that we have only been able to secure a single reviewer to assess your manuscript. We are issuing a decision on your manuscript at this point to prevent further delays in the evaluation of your manuscript. Please be aware that the editor who handles your revised manuscript might find it necessary to invite additional reviewers to assess this work once the revised manuscript is submitted. However, we will aim to proceed on the basis of this single review if possible. The reviewer's comments are positive but they have identified a number of opportunities to improve the manuscript. Please pay particular attention to addressing their queries regarding the methods, so as to ensure that the manuscript satisfied PLOS ONE's third publication criterion (https://journals.plos.org/plosone/s/criteria-for-publication).

We look forward to receiving your revised manuscript.

Kind regards,

Jamie Males

Editorial Office

PLOS ONE

Journal Requirements:

"This work was financially supported by a research grant from the Social Insurance Agency [grant number 059159-2015]. We utilised data from the REWHARD consortium supported by the Swedish Research Council (grant number 2017-00624)."

Reviewers' comments:

Reviewer's Responses to Questions

**Comments to the Author**

1. Is the manuscript technically sound, and do the data support the conclusions?

Reviewer #1: Yes

2. Has the statistical analysis been performed appropriately and rigorously? 

Reviewer #1: Yes

3. Have the authors made all data underlying the findings in their manuscript fully available?

Reviewer #1: Yes

4. Is the manuscript presented in an intelligible fashion and written in standard English?

Reviewer #1: Yes

5. Review Comments to the Author

Reviewer #1: Overall, this is a well-written study presenting a prognostic model for predicting the duration of 20,049 sickness absence spells due to shoulder lesions in a population-based cohort in Sweden. The authors conclude that age, sex, geographical region, occupational status, educational level, birth country, specialized healthcare at start of the spell, number of sickness absence days in the last 12 months, and specialized healthcare in the last 12 months predict duration of sickness absence. The model had limited predictive power, but could discriminate the very long sickness absence spells from the rest.

This is a solidly performed study, and I have focused my feedback mainly on the interpretation of the study results:

1. In the introduction the authors mention that they will focus this study on the clinical implementation. Therefore, they select a relatively small number of predictors. However, all these predictors are unchangeable, and do not necessarily it in clinical practice. Can the authors clarify their rationale for focusing this model on clinical practice, in the introduction section?

2. Following up on the first question: Can the authors include implementations for (clinical) practice in the discussion section?Who can use the results of this study? Is this meant for further research, or to inform policy-makers, occupational health professionals, or clinicians?

3. The conclusion states: '.......and may also be used to identify patients groups at higher risk of very long SA for ease of directing rehabilitation resources'. If this is your main conclusion, the results should be more focused on predicting the long SA spells.

Questions related to the methodology:

1. What was the rationale for only including SA spells that lasted > 14 days?

2. Why was data between 2010-2012 used? Would this have any implications for translating the results to 2022? Can you discuss this, and mention it as a limitation of the study?

3. Can the authors specify how the variable selection of the 14 predictors was performed?

4. Have the authors done a sensitivity analysis in which they separated the individuals who had more than 1 sickness absence spell? It is known that workers who have more than one sickness absence spell have a very different sickness absence course.

Minor comments:

1. In the abstract the predictors don't give direction. Instead of presenting 'gender' as a predictor, can the authors present 'being a man' or 'being a woman' as a predictor

2. Can the authors present the in-and exclusion criteria, and a flow-chart presenting the selection procedure of individuals and sickness absence spells?

3. The first sentence of the discussion ('We could arrive at...') is unclear. Can the authors please rewrite this sentence?

6. PLOS authors have the option to publish the peer review history of their article (what does this mean?). If published, this will include your full peer review and any attached files.

Reviewer #1: No

---

## [Author Response · Author response to Decision Letter 0]

27 Sep 2022

Manuscript number PONE-D-22-17574N

Dear Editor,

Thank you very much for the opportunity to submit a revised version of our manuscript entitled “A prognostic model for predicting the duration of 20,049 sickness absence spells due to shoulder lesions in a population-based cohort in Sweden”. We appreciate the reviewer’s valuable comments, and have now revised the manuscript according to them, using track changes. Below we address each point of the reviewer’s comments. We also made some minor language editing.

Regarding the Editor’s comments, we have now updated the Data Availability Statement in line with your requirements as following: The data used in this study is administered by the Division of Insurance Medicine, Karolinska Institutet, and cannot be made public. According to the General Data Protection Regulation, the Swedish law SFS 2018:218, the Swedish Data Protection Act, the Swedish Ethical Review Act, and the Public Access to Information and Secrecy Act, these types of sensitive data can only be made available, after legal review, for researchers who meet the criteria for access to this type of sensitive and confidential data. Readers may contact Professor and Head of Division Ellenor Mittendorfer-Rutz (ellenor.mittendorfer-rutz@ki.se) regarding the data. 

Regarding funding statement, it should be revised, in line with your requirements to: "This work was financially supported by a research grant from the Social Insurance Agency [grant number 059159-2015, receiver: KA]. We utilised data from the REWHARD consortium supported by the Swedish Research Council (grant number 2017-00624, receiver: KA). There was no additional external funding received for this study. The funders had no role in study design, data collection and analysis, decision to publish, or preparation of the manuscript."

Now we also included captions for the Supporting information file to manuscript and updated in-text citations accordingly.

Yours sincerely, on behalf of all authors,

Katalin Gemes

Point-by-point responses to reviewer comments

Reviewer #1:

Overall, this is a well-written study presenting a prognostic model for predicting the duration of 20,049 sickness absence spells due to shoulder lesions in a population-based cohort in Sweden. The authors conclude that age, sex, geographical region, occupational status, educational level, birth country, specialized healthcare at start of the spell, number of sickness absence days in the last 12 months, and specialized healthcare in the last 12 months predict duration of sickness absence. The model had limited predictive power, but could discriminate the very long sickness absence spells from the rest.

This is a solidly performed study, and I have focused my feedback mainly on the interpretation of the study results:

1. In the introduction the authors mention that they will focus this study on the clinical implementation. Therefore, they select a relatively small number of predictors. However, all these predictors are unchangeable, and do not necessarily it in clinical practice. Can the authors clarify their rationale for focusing this model on clinical practice, in the introduction section?”

Author’s response: Thank you for this comment. You are right, these predictors concern the situation for the patient at the beginning of a new SA spell. In this study we have no information on different types of treatments, nor of aspects that might change in the future – the latter is usually the case in predictions. General practitioners expressed that they, at least, want information on possible basic predictors. Even if we wanted to include more possible predictors in the model, the aim of this study was to include a very limited number, in order to ensure that it would be used in primary healthcare. We have now included more information about this in the Introduction section, in line with your comment.

(p 4 line 22-24). “that is, studies based on large, non-selected data are needed. Physicians in primary healthcare, that is, general practitioners, have asked for a soft-ware based tool regarding such predictions, not requiring much work or time to use.” 

2.” Following up on the first question: Can the authors include implementations for (clinical) practice in the discussion section? Who can use the results of this study? Is this meant for further research, or to inform policy-makers, occupational health professionals, or clinicians?”

Author’s response: Our model is mainly meant to be used in primary health care practice, by sickness certifying general practitioners. The use of the model has now been introduced at some such practices. We hope our study will inspire future studies regarding predictive models, predictors, and follow-ups of the use of the model. We added the following sentences to the Discussion (p 12-13 and line 23-25, 1-6): “This prediction model can be used in Sweden, during primary healthcare consultations when patients begin a SA spell due to shoulder lesion. The general practitioner inserts information on the relevant predictors about the patient and optionally specifies the length of SA for which to obtain a probability. The model outputs a probability score that can be used by the practitioner to early identify patients with high risk for long-term SA. The model’s output can also be used as a basis for discussion with the patient regarding their current SA. Early identification of those SA spells that are at high risk of becoming long means that resources and support, such as rehabilitation measures, can more accurately be targeted to those with greater need of them.” 

”3. The conclusion states: '.......and may also be used to identify patients groups at higher risk of very long SA for ease of directing rehabilitation resources'. If this is your main conclusion, the results should be more focused on predicting the long SA spells.”

Author’s response: You are right, this is one of our main conclusions from this study, and we have now revised the text about this in the Conclusion section. The conclusion text now reads: “A prediction model of the duration of sickness absence (SA) spells due to shoulder lesion was developed, with ease of clinical implementation in mind. Discriminatory ability was poor in the short term but improved for predicting very long durations of SA. The model is useful for prognostications of duration of SA spells and can provide information regarding which cases that might be in need of extra rehabilitation measures to promote return to work or other activity. However, future external validation is recommended.”

” Questions related to the methodology:

1. What was the rationale for only including SA spells that lasted > 14 days?”

Author’s response: The reason for this was very pragmatic: we based this study on data from registers of the Swedish Social Insurance Agency. We from those registers had access to information about all SA spells that lasted >14 days. Benefits for the first 14 days of a SA spell is provided by the employer or by the Social Insurance Agency for those unemployed or self-employed. In order not to introduce bias we only used SA spells >14 days. Moreover, a physician certificate regarding the diagnoses leading to the need of SA is required for these longer SA spells, why the validity of the diagnoses is good for these longer spells. Using these registers, we could include all new such SA spells due to shoulder lesions >14 days that occurred in Sweden in the inclusion period. We added the following sentence to the Methods (page 6, lines 6-7): “In order not to introduce bias regarding unemployment, we only included SA spells >14 days. This also led to high validity of SA diagnoses.”

”2. Why was data between 2010-2012 used? Would this have any implications for translating the results to 2022? Can you discuss this, and mention it as a limitation of the study?”

Author’s response: When this project started, data from 2010-2012 were the most recent complete data available for the analyses, which allowed long enough follow-up of each SA spells till its end. We, of course, agree that other studies, based on later data are needed, to study if our results are still relevant. These types of predictive models need to be updated regularly when new data is available. However, no significant legislative changes regarding SA benefits have occurred since then, why it is likely that the results can also apply to 2022. We discuss this limitation on page 13 (line 20-24) as following: “Furthermore, our results concern SA spells initiated in 2010-2012, but as not major legislative changes regarding SA benefit have been implemented in 2012-2020, the results are likely to be valid to later periods. However, the predictive models need to be updated to examine possible needs of changing predictors - when new data is available to ensure its validity during later periods.” 

3. Can the authors specify how the variable selection of the 14 predictors was performed?

Author’s response: There were several criteria we considered in the pre-selection of the 14 variables: we selected variables based on previous knowledge about their possible influence on the duration of SA. Many of these variables were, as stated in the manuscript, redundant and/or highly correlated, e,g., net and gross days of SA or disability pension spells, or referred to different periods, as during 1-365 days, 366-730, or 1-730 days before the start date of the index SA spell (that is, what happened in the first year before the start date, in the second year before the start, or in the two years combined). We checked their predictive value, especially comparing variables referring to specific diagnoses vs. their non-diagnosis specific pair. One of the selection criteria was between the global and the diagnosis-specific variables (that is, information on, e.g., previous SA, DP, or healthcare due to specific diagnoses, compared to information on all previous SA, DP, or healthcare, respectively.) If they had the same predictive value (in most cases the global one had even better predictive value), the global variable was kept, as the model was intended to be used in clinical practice and some information for its application was to be provided by the patient during the consultation. Variables that were easy to remember and could be accurately assessed were prioritized. For example, the number of SA days in the preceding year before the start date of the index SA spell was chosen over the diagnosis-specific number of SA days in the preceding year. Similarly, if there was no difference in whether the variable was assessed for two or for one years prior the start of the index spell, then the shorter period was chosen to facilitate implementation. The remaining 14 predictors were generally available from a patient-physician consultation and were independent predictors capturing different aspects that may influence future SA. After discussions with clinicians, we fore the final model selected the six variables with the highest predictive values. We now have added a more detailed specification of the selection of the set of 14 variables to the manuscript at page 6-7 and the modified paragraph now reads as following: “A priori, it was decided to develop a final model with no more than nine predictors, including age, sex, and geographic region, which can be easily and reliably assessed in a short time period during a primary healthcare visit to promote implementation of the model into clinical practice. Initially, around 130 predictors were considered to be included in the model. In order to decrease the number of predictors to a reasonable number that could be used to build the prediction model, several parallel preselection processes were applied, based on feasibility for clinical implementation and scope for association to the outcome. The majority of the initial predictors were redundant or partially redundant and/or highly collinear with at least one other predictor, e.g., net and gross days of the same measures in days. Therefore, based on analyses results, a subset of them were selected for further consideration. For predictors that were hierarchically related, e.g., multi-morbidity and cause-specific morbidity, general and cause-specific hospitalization, etc., choosing only the general/coarse variables over the specific ones would improve both parsimony and the scope for implementation of the model in clinical settings and was, therefore, the selected strategy. This variable selection procedure resulted in a set of 14 easily available and easy to assess variables, with minimal collinearity and strong associations with the outcome for modeling”

“4. Have the authors done a sensitivity analysis in which they separated the individuals who had more than 1 sickness absence spell? It is known that workers who have more than one sickness absence spell have a very different sickness absence course.”

Author’s response: We did not consider sensitivity analysis, as the model was developed for sickness absence (SA) spells not for individuals and at the beginning of the index spell the length of the SA spell was not known, nor if they would have future other SA spells. A sensitivity analysis would be able to use the second spells that started within the study period, but not those that started outside the study period. Moreover, you are right that people with previous SA have a higher risk of future SA, which is the main reason for why we included information on previous SA in the 12 months before the date of the beginning of the index SA spell as a possible predictor to our model. For the few people in this study who had a new SA spell that was included, the previous included SA spell would be included as a possible predictor if it occurred within 12 months. When including SA spells for the previous 2 years did, as stated above, not increase the predictive value. 

Minor comments:

1. In the abstract the predictors don't give direction. Instead of presenting 'gender' as a predictor, can the authors present 'being a man' or 'being a woman' as a predictor”

Author’s response: Actually, focus in this study is on the development and validation of the model, that can be used to predict duration of a new SA spell and not the directions of the associations between specific predictors and the outcome. In that sentence we rather described what predictors that were included in the model (e.g., age, sex etc), than the specific values of them. However, we had information on sex, not gender, and have now revised this in the abstract. 

”2. Can the authors present the in-and exclusion criteria, and a flow-chart presenting the selection procedure of individuals and sickness absence spells?”

Author’s response: Actually, all the SA spells fulfilling the inclusion criteria remained in the analyses, none were excluded, why we suggest a flow chart is not necessary. However, we now rephrased the Methods slightly to make the inclusion criteria clearer. 

”3. The first sentence of the discussion ('We could arrive at...') is unclear. Can the authors please rewrite this sentence?”

Author’s response: This sentence has now been revised as following:” We developed a prediction model that could be implemented in primary healthcare”.

---

## [Decision Letter · Decision Letter 1]

6 Dec 2022

PONE-D-22-17574R1A prognostic model for predicting the duration of 20,049 sickness absence spells due to shoulder lesions in a population-based cohort in SwedenPLOS ONE

Dear Dr. Gemes,

Thank you for submitting your manuscript to PLOS ONE. After careful consideration, we feel that it has merit but does not fully meet PLOS ONE’s publication criteria as it currently stands. Therefore, we invite you to submit a revised version of the manuscript that addresses the points raised during the review process. Reviewer One had no further comments to your manuscript but reviewer Two had some minor comments that I would like you to handle in the Discussion part in the paper.1. Please discuss the handling of the occupational codes.2. Have you more detailed information about the diagnosis than M75? Probably not, but please comment this in the Discussion part.3. Please discuss the sick leave guidelines from the National Board of Helth and Welfare, and if those guidelines may have affected the length of the sick leaves.

We look forward to receiving your revised manuscript.

Kind regards,

Kjell Torén, MD, PhD

Academic Editor

PLOS ONE

Journal Requirements:

Reviewers' comments:

Reviewer's Responses to Questions

**Comments to the Author**

1. If the authors have adequately addressed your comments raised in a previous round of review and you feel that this manuscript is now acceptable for publication, you may indicate that here to bypass the “Comments to the Author” section, enter your conflict of interest statement in the “Confidential to Editor” section, and submit your "Accept" recommendation.

Reviewer #1: All comments have been addressed

Reviewer #2: (No Response)

2. Is the manuscript technically sound, and do the data support the conclusions?

Reviewer #1: Yes

Reviewer #2: Yes

3. Has the statistical analysis been performed appropriately and rigorously? 

Reviewer #1: Yes

Reviewer #2: Yes

4. Have the authors made all data underlying the findings in their manuscript fully available?

Reviewer #1: Yes

Reviewer #2: Yes

5. Is the manuscript presented in an intelligible fashion and written in standard English?

Reviewer #1: Yes

Reviewer #2: Yes

6. Review Comments to the Author

Reviewer #1: All comments have been sufficiently addressed.

The manuscript is technically sound, the statistical analyses have been performed appropriately, an appropriate data-sharing statement has been given, and the manuscript is written in standard English.

Reviewer #2: Thank you for the opportunity to review this interesting and timely paper from Gemes at al. It is a nice study dealing with a prognostic model for predicting the duration of 20,049 sickness absence spells due to shoulder lesions in a population-based cohort in Sweden. A prediction model of the duration of sickness absence (SA) spells due to shoulder lesion was developed, with ease of clinical implementation in mind. Discriminatory ability was poor in the short term but improved for predicting very long durations of SA. The model could be useful for prognostications of duration of SA spells and can provide information regarding which cases that might be in need of extra rehabilitation measures to promote return to work.

The study is solidly performed, and the topic is relevant, and as the authors point out, there is currently lack of scientific knowledge in the area.

There are some areas that require attention, and these are noted below:

1. Work ability depends on the individual's resources but also on the physical and psychosocial demands of the work. The LISA database provides access to occupational codes. Have the authors included, for example, job sector or occupation as a predictor? If not, I suggest that the authors discuss why and raise this in the discussion section.

2. The authors have used ICD10-code M.75, and as they describe, they do not have access to more detailed information with the subgroups that exists. Regarding sickness absence, it is for example probably more likely that a person with adhesive capsulitis has a longer sickness absence than some of the other diagnoses in this group. I suggest that the authors clarifying that problem in the discussion section.

3. In Sweden, the National Board of Health and Welfare has a sick leave support for shoulder problems, which is there to help the physicians to predict how long a person with a certain diagnosis is recommended to be on sick leave. Could this have influenced the outcome and be a predictor for how long the people were on sick leave? It would be helpful if this also was raised in the discussion section.

7. PLOS authors have the option to publish the peer review history of their article (what does this mean?). If published, this will include your full peer review and any attached files.

Reviewer #1: No

Reviewer #2: No

---

## [Author Response · Author response to Decision Letter 1]

19 Dec 2022

Manuscript number PONE-D-22-17574R1

Title: A prognostic model for predicting the duration of 20,049 sickness absence spells due to shoulder lesions in a population-based cohort in Sweden

Dear Editor Kjell Torén, 

Thank you very much for the opportunity to submit a revised version of this manuscript! We appreciate reviewer #2’s additional valuable comments as well as the comments from you and have addressed them point by point below. We have also revised the manuscript accordingly, using the Track-Changes tool.

On behalf of all authors, yours sincerely,

Katalin Gemes

Comments from reviewer #2

Thank you for the opportunity to review this interesting and timely paper from Gemes at al. It is a nice study dealing with a prognostic model for predicting the duration of 20,049 sickness absence spells due to shoulder lesions in a population-based cohort in Sweden. A prediction model of the duration of sickness absence (SA) spells due to shoulder lesion was developed, with ease of clinical implementation in mind. Discriminatory ability was poor in the short term but improved for predicting very long durations of SA. The model could be useful for prognostications of duration of SA spells and can provide information regarding which cases that might be in need of extra rehabilitation measures to promote return to work.

The study is solidly performed, and the topic is relevant, and as the authors point out, there is currently lack of scientific knowledge in the area.

Author’s response: Thank you for these positive comments.

There are some areas that require attention, and these are noted below:

 “1. Work ability depends on the individual's resources but also on the physical and psychosocial demands of the work. The LISA database provides access to occupational codes. Have the authors included, for example, job sector or occupation as a predictor? If not, I suggest that the authors discuss why and raise this in the discussion section.”

Editor’s comment: “Please discuss the handling of the occupational codes.”

Author’s Response: Different occupational aspects were considered at the preselection phase as possible predictors. Those were: type of work (white or blue collar), type of work sector (private, public), and work situation at the start of the SA spell (in paid work, on parental leave, studying, unemployed) – of those three, only the last one had strong enough predictive value to be included in the analyses. This is not what we had expected- as your comment suggests, one would have expected that at least blue/white collar would be predictive – why we of course checked those analyses extra many times. Due to these results, we did not study specific occupations. 

Probably the type of occupation has more impact on the risk of becoming sickness absent; not on the duration of SA spells among those who actually are on SA. The same goes e.g., for sex; women have a higher risk for becoming sickness absent, however, among those sickness absent, women in general have the same risk of long-term SA spells.

We now added more information about this in the Method section: 

(Some of the variables that in the initial process did not show strong enough predictive value to be included among those 14, were two of the work related: namely type of sector /private vs. public) and type of work (blue vs. white collar)). (page 7)

Reviewer’s comment 2: “The authors have used ICD10-code M.75, and as they describe, they do not have access to more detailed information with the subgroups that exists. Regarding sickness absence, it is for example probably more likely that a person with adhesive capsulitis has a longer sickness absence than some of the other diagnoses in this group. I suggest that the authors clarifying that problem in the discussion section.”

Editor’s comment: 2. “Have you more detailed information about the diagnosis than M75? Probably not, but please comment this in the Discussion part.”

Author’s response: We, of course, agree with the reviewer and Editor that the future work capacity as well as the SA prognosis might differ between M75 sub diagnoses within this group, as it would do with level of severity and/or type of treatment within each of those sub groups. Unfortunately, we didn’t have more detailed information on the SA diagnosis, nor on disease severity or on type of treatments. We now revised the limitation text as follows:

We had no information on disease severity, nor on type of treatment (e.g., surgery or not), nor information on specific SA diagnostic M75 sub groups (we only had information about the three-digit level ICD-10 code from MiDAS). Such types of information could have improved the discriminatory ability of the model and would be preferable to include in future studies [6]. (page 12)

Reviewer’s comment: 3. “In Sweden, the National Board of Health and Welfare has a sick leave support for shoulder problems, which is there to help the physicians to predict how long a person with a certain diagnosis is recommended to be on sick leave. Could this have influenced the outcome and be a predictor for how long the people were on sick leave? It would be helpful if this also was raised in the discussion section.”

Editor’s comment 3: “Please discuss the sick leave guidelines from the National Board of Helth and Welfare, and if those guidelines may have affected the length of the sick leaves.”

Authors’ response: There are, of course, many aspects that influence the duration of a SA spell. In Sweden, in the last decades, a large number of nationwide interventions were introduced in order to influence SA, due to the very high SA rates at the turn of the century. Some of those that had been introduced at the time of these SA spells are: interventions to increase to competence in insurance medicine among physicians and among social insurance officers, competence developments of healthcare CEOs regarding their management of how sickness certification of patients is handled in their clinic, stricter time limits regarding when a SA spell is to be checked by the social insurance officer (those are obvious in Figure 1, that is, at day 90, day 180, 365, and 914 of a SA spell), that social insurance officers to a larger extent also did check this (that is, a change in practice), more scientific knowledge about possible ‘side affects’ of being sickness absent among both claimants and professionals. Another of those interventions was the one you mention; that the National Board of Health and Welfare’s in 2007 introduced nationwide general recommendations for handling of sickness certification and for possible duration of sickness absence due to specific diagnoses. 

The recommendations ‘only’ cover the most common SA diagnoses, and thus not all sub groups of M75. Those four subgroups that were covered at this time, all together under the same headline in the recommendations, were M75.0 (frozen shoulder), M75.1 (rotator cuff injuries), M75.3 (calcific tendinitis), and M75.4 (impingement). The recommendations stated that:

-M75.0; up to 2 weeks, if physically demanding job up to three weeks. After surgery: return to work (RTW) within a few days, of not physically demanding work. Otherwise, physically demanding job SA up through a month.

-M75.1. up to six weeks, if physically very demanding, up to 26 weeks.

-M75.3: up to 2 weeks, if physically demanding job up to three weeks.

-M75.4: up to 2 weeks, if physically demanding job up to three weeks. After surgery, SA up to three weeks if not physically demanding job, if physically very demanding job up to 12 weeks.

Moreover, in those recommendations it is clearly stated that the need for SA is to be individually assessed, and can vary widely for people with the same diagnoses. For instance, for the M75 diagnoses, the need might be shorter if not the dominant arm is affected. However, other aspects, such as multimorbidity, might lead to need of longer duration. Altogether, less than 20% of the M75 SA spells exceed the longest recommended time (26 weeks).

It is, of course, an interesting study question whether those recommendations influenced the duration of SA spells due to M75, however, such a study would have required both other data and another study design. Moreover, it would not be easy to differentiate effects from those recommendations from effects of the other interventions. 

We have now, nevertheless, included a paragraph about this at the end of the Discussion section and also added two references regarding the recommendations for SA duration.

In Sweden, several nationwide interventions have been introduced in the decade before those SA spells begun, such as interventions to increase competence in insurance medicine among physicians and social insurance staff, stricter time lines regarding when to assess claimants right to prolonged the SA spell (e.g., at day 90, 180, 365, and 914, as can be seen in Figure 1), and in 2007, recommendations from the Board of Health and Welfare regarding sickness certification and duration of SA spells due to specific diagnoses, including M75, namely due to M75.0 (frozen shoulder), M75.1 (rotator cuff injuries), M75.3 (calcific tendinitis), and M75.4 (impingement) [35, 36]. The degree to which those interventions might have had an impact on how long the SA spells became would require other types of studies and data. (page 13)

---

## [Editor Report · Decision Letter 2]

20 Dec 2022

A prognostic model for predicting the duration of 20,049 sickness absence spells due to shoulder lesions in a population-based cohort in Sweden

PONE-D-22-17574R2

Dear Dr. Gemes,

We’re pleased to inform you that your manuscript has been judged scientifically suitable for publication and will be formally accepted for publication once it meets all outstanding technical requirements.

Kind regards,

Kjell Torén, MD, PhD

Academic Editor

PLOS ONE

Additional Editor Comments (optional):

I think the additional changes you have presented make the manuscript suitable for publication.

Accept!
---

## [Editor Report · Acceptance letter]

3 Jan 2023

PONE-D-22-17574R2 

A prognostic model for predicting the duration of 20,049 sickness absence spells due to shoulder lesions in a population-based cohort in Sweden 

Dear Dr. Gémes:

I'm pleased to inform you that your manuscript has been deemed suitable for publication in PLOS ONE. Congratulations! Your manuscript is now with our production department. 

Kind regards, 

on behalf of

Dr. Kjell Torén 

Academic Editor

PLOS ONE